# POLFORMER: EVENT-ONLY SELF-SUPERVISION WITH PROBABILISTIC ATTENTION FOR ROAD SEGMENTATION

## ABSTRACT

Event cameras offer **microsecond latency and exceptional dynamic range**, making them a natural fit for road segmentation in autonomous driving. Yet their impact has been limited by the scarce annotations and the high cost of labeling event streams. Current solutions rely on transferring knowledge from RGB domains, but this dependence erases the very advantages that make event sensing unique.

This work break the dependence on RGB with an **event-native self-supervised transformer architecture** that learns rich event-specific semantics directly from raw unlabeled event streams (no frames) through a polarity-guided self-supervised pretext task. To further exploit the spatiotemporal richness of event data, we propose a **probabilistic attention mechanism** that outperforms standard dot-product attention on this modality.

On DSEC-Semantic and DDD17, our approach achieves state-of-the-art road segmentation with **orders of magnitude fewer labels**. These results establish self-supervision as a scalable and label-efficient paradigm shift for event-driven vision in autonomous driving.

.

## 1 INTRODUCTION

Autonomous driving depends critically on reliable road segmentation (1), yet frame-based pipelines remain mismatched to real-time operation due to dense, synchronous image streams that inflate latency and compute budgets (2) (3) (5). Event cameras (6) provide sparse, asynchronous measurements with microsecond latency by firing only on brightness changes, making them naturally suited to efficient, low-latency perception under tight compute and power constraints.

Despite this promise, event-based segmentation lags because labels are scarce and many methods attempt to retrofit events into frame-like grids and transferring supervision from RGB, thereby sacrificing sparsity and entangling learning with frame-domain annotations that scale poorly. The bottleneck is acute: labeling events requires sub-millisecond, fine-grained annotation that is far more labor-intensive than pixel-level image labeling, impeding the creation of large supervised corpora.

We address this challenge with **PolFormer, a Polarity-driven Self-Supervised Transformer**, that learns rich event-specific representations directly from raw event streams. By leveraging polarity as an intrinsic supervisory signal, PolFormer enables scalable pretraining without frame labels. Complementing this, we design a **probabilistic attention** module that explicitly models spatial locality between events, yielding stronger representations than dot-product attention. Unlike conventional dot-product attention, our approach integrates spatial distance to model the probability of interaction between events.

### 1.1 CONTRIBUTIONS

Our main contributions are:

1. **1. Event-only self-supervision**: A self-supervised, event-only pretraining framework that leverages polarity to learn semantically meaningful features without any frame-domain labels, targeting the core data-scarcity barrier in event segmentation.

2. **2. Probabilistic attention** A novel attention mechanism that combines query–key similarity with a spatial-distance likelihood, improving representation quality under sparse supervision.

3. **3. Label efficiency and Transferability**: Demonstration on DSEC-Semantic and DDD17 that PolFormer achieves state-of-the-art road segmentation with drastically fewer labels, while transferring effectively across datasets.

## 2  RELATED WORK

This section provides an in-depth survey of two related domains of road segmentation: semantic segmentation and self-supervised learning using an event camera.

### 2.1  SEMANTIC SEGMENTATION IN EVENT DOMAIN

The first event-based semantic segmentation baseline, Ev-SegNet (7), introduced a 6-channel event representation with an Xception-based CNN (9) and extended the DDD17 dataset (8) for segmentation. Later works expanded training data via synthetic events (10), or transferred supervision through knowledge distillation (13). Multi-modal frameworks such as ISSAFE (14) and ESS (15) combined events with RGB, while transformer-based designs with posterior attention (25) advanced architectural choices. Other efforts targeted latency with multi-latent memories (40) or multimodal fusion (e.g., OpenESS (39)), but these still inherit annotation costs and domain biases from RGB.

### 2.2  SELF SUPERVISED LEARNING IN EVENT DOMAIN

The absence of labelled event data in a number of vision tasks such as optical flow (20), intensity reconstruction (21), object classification (22) *etc.*, has been addressed by SSL (17) (18) (19). These works either rely on massive amounts of RGB data or synchronous, pixel-aligned RGB and event data recording. (23) (15) (13) transfer knowledge from the RGB domain to the event domain with unpaired labelled frames and unlabeled events. These methods rely on data acquired under similar conditions from both frame-based and event-based cameras, which is often not feasible. Masked Event Modeling (MEM), an SSL framework dependant only on event data, was proposed (16). It employs partially removed event data reconstruction as a pre-task. EV-LayerSegNet (41) proposed introduced a self-supervised CNN for event-based motion segmentation.

### 2.3  OUR UNIQUE POSITIONING

To summarize, existing approaches either (i) aggregate events into frame-like structures, sacrificing sparsity and asynchrony, or (ii) rely on RGB supervision, binding event-based progress to frame-domain labels. More recent event-only SSL frameworks such as Masked Event Modeling (MEM) attempt to bypass this by reconstructing masked event streams. However, MEM primarily encourages recovery of low-level firing statistics and does not guarantee that learned features align with semantics required for downstream tasks like road segmentation.

PolFormer departs from both trends. By leveraging polarity entropy as an intrinsic supervisory signal, PolFormer learns task-aligned, semantically meaningful representations that directly capture object boundaries and motion cues. In addition, our probabilistic attention mechanism explicitly models spatial locality in event streams, further strengthening representation quality. Together, these advances establish PolFormer as the novel event-only self-supervised transformer designed specifically for segmentation, achieving strong label efficiency and cross-dataset transfer while fully preserving the native benefits of event sensing.

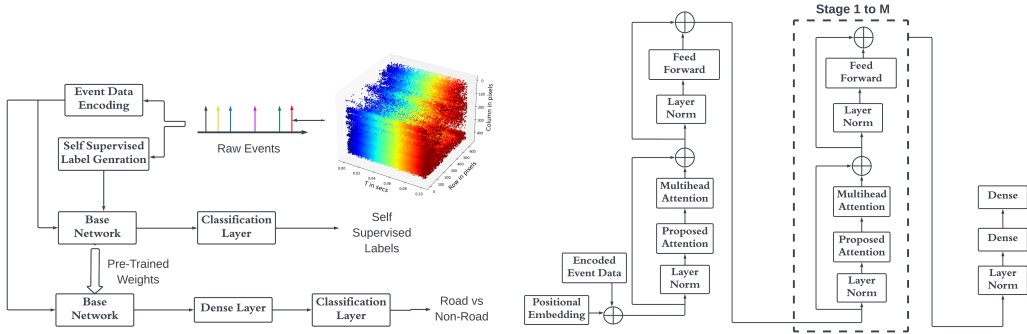

(a) Self-Supervised PolFormer          (b) Base Network of PolFormer

Figure 1: (a): Framework of PolFormer for self-supervised learning of pre-trained weights. Self-supervised labels are generated from the polarity of raw events. A base network and a classification layer are trained to minimize the cross-entropy loss between true and predicted self-supervised labels. Once training converges, the classification layer is replaced with a small feed-forward neural network with a dense layer of 128 neurons and a classification layer. The network is fine-tuned with a few labelled samples of the downstream road segmentation task. (b) Base network of PolFormer. Transformer architecture with four feature extraction blocks. Each block comprises layer normalization, residual connection, multi-head attention with the proposed attention mechanism and a feed-forward neural network.

## 3 PROPOSED SOLUTION

The central challenge in event-based road segmentation is to learn semantically meaningful representations directly from raw event streams, without relying on frame conversions or RGB supervision. To address this, we design PolFormer, an event-native transformer tailored to the sparse, asynchronous structure of events. Our solution builds on two key ideas:

**1. Polarity-driven self-supervision**: Event polarity encodes whether brightness increases or decreases, providing a natural supervisory signal that reflects object boundaries and motion patterns. By framing a self-supervised pretext task around polarity entropy, we enable scalable pretraining from unlabeled event streams. This yields features aligned with downstream segmentation tasks, even under label scarcity.

**2. Probabilistic attention**: Conventional dot-product attention ignores spatial structure, treating all event interactions uniformly. In contrast, we introduce a probabilistic attention mechanism that combines query–key similarity with a spatial-distance likelihood. This explicitly favors local, semantically relevant event interactions, improving both efficiency and representation quality.

The resulting architecture (Fig. 1) consists of a self-supervised transformer backbone pretrained on raw events, followed by task-specific layers fine-tuned with limited labeled data. This design ensures that PolFormer remains event-native throughout: it does not project events into frames, does not depend on RGB annotations, and preserves the computational and latency advantages of event sensing.

### 3.1 EVENT TRANSFORMER ARCHITECTURE

The proposed architecture starts with an input tensor $E = \{e_1, e_2 \dots e_N\}$ of dimension $N \times 4$, where $N$ represents the number of events and $e_i = (x_i, y_i, t_i, p_i)$, where $p \in [+1, -1]$ represents increase or decrease in brightness respectively. A fully connected layer projects the input tensor onto vectors of dimension 12. Positional embeddings have been added via the standard positional embedding layer. Subsequently, the architecture incorporates four feature extraction blocks, each of which consists of a multi-head attention and a feed-forward neural network. Multi-head attention employs 4 heads with the proposed probabilistic attention mechanism. The output of multiple heads is combined later. The feed-forward neural network includes two dense layers with 24 and 12

nodes with GeLU activation. To ensure stability and accelerate convergence, layer normalization and residual connections are integrated. The feature extraction is followed by a feed-forward neural network with 2048 and 1024 neurons with GeLU activation. This base architecture is appended with task-specific layers.

## 3.2 SELF SUPERVISED LEARNING FRAMEWORK

Transformer networks are typically trained using data paired with their corresponding supervisory labels. In the conventional vision, state-of-the-art transformer networks are learnt from a huge corpus of supervisory images. Unfortunately, being an emerging field, the event camera domain suffers from a lack of sufficient labelled data pertaining to specific tasks. Hence, to obtain better performance with limited supervisory signals, the proposed approach investigates the utility of the information embedded in the polarity of the event sequence as a supervisory signal.

### 3.2.1 INTUITION

Polarity indicates whether a pixel's brightness increases (+1) or decreases (–1). The distribution of polarities in a sequence reflects scene dynamics: moving objects, edges, and motion cues all create characteristic polarity patterns. By designing a pretext task around polarity entropy, we train the model to distinguish structured, dynamic regions from random noise. This allows PolFormer to learn semantically aligned representations without a single frame-domain label.

### 3.2.2 FRAMEWORK

Given an unlabeled training dataset $E = \{e_i\}_{i=1}^N$ of $N$ events, the distribution of polarity is modelled as binomial distribution, and the probability of occurrence of positive events is estimated as

$$p^+ = \sum_{i=1} N \mathbb{I}(p_i == 1) \tag{1}$$

The entropy of the event sequence $E$ is estimated as follows,

$$H = -p^+ \log p^+ - (1 - p^+) \log(1 - p^+) \tag{2}$$

The supervisory signal $y_i^s$ for each event sequence $E_i$ is formed as follows,

$$y_i^s = \begin{cases} 1, & \text{if } H > a \\ 0, & \text{otherwise} \end{cases} \tag{3}$$

High entropy indicates diverse motion dynamics, while low entropy corresponds to uniform or static regions. This entropy is converted into a binary supervisory label. The network is trained to minimize the cross entropy loss between predicted $\hat{y}_i^s$ (softmax of the network output) and true label $y_i^s$. The objective is simple cross-entropy loss, but the effect is powerful: the network is forced to capture meaningful spatial–temporal structure in events to solve the task.

Once pretrained, the classification head is replaced with a lightweight segmentation head, and the backbone is fine-tuned on a small set of labeled road segmentation samples. Because pretraining already encodes motion boundaries and polarity dynamics, the model fine-tunes effectively with very few labels.

## 3.3 PROBABILISTIC ATTENTION

The general attention mechanism followed in Transformer architecture (28). The given input sequence $E = [e_1, e_2 \ldots e_N] \in R^{N \times d_e}$ is projected into query $(Q \in R^{N \times d})$, key $(K \in R^{N \times d})$ and value $(V \in R^{N \times d})$ matrices through the following transformation,

$$Q = EW_q^T; K = EW_k^T; V = EW_v^T \tag{4}$$

Where, $W_q, W_k$ and $W_v \in R^{d \times d_e}$ are the weight matrices, $Q = [q_1, q_2 \ldots q_N]$, $K = [k_1, k_2 \ldots k_N]$ and $V = [v_1, v_2 \ldots v_N]$ and $q_i$, $k_i$ and $v_i$ are query, key and value vectors respectively.

The $i^{th}$ output is estimated as the weighted average of the value vectors, $o_i = \sum_{j=1}^{N} w_{ij} v_j$, where the attention scores $w_{ij}$ are generally estimated as scaled dot product attention as follows,

$$softmax \left( \frac{q_i^T k_j}{\sqrt{d}} \right) \tag{5}$$

Standard transformer attention computes similarity using a scaled dot product between query and key vectors. While effective in frame-based vision, this formulation ignores the spatial sparsity and locality that define event data: events occur at discrete pixels and are strongly correlated with nearby events in both space and time. Treating all pairwise interactions uniformly dilutes these structural cues.

To address this, we propose a probabilistic attention mechanism that augments query–key similarity with a spatial likelihood term. Intuitively: i) The **first term** measures how likely a key vector is to correspond to a given query, following a probabilistic similarity model. ii) The **second term** encodes the probability of observing another event at a certain spatial distance from the current one, capturing locality.

The final attention score combines these two probabilities, ensuring that interactions between spatially relevant events are emphasized, while far-apart or weakly related events are downweighted. Formally, given queries $q_i$ and keys $k_j$, we define the attention weight as:

$$w_{ij} = \mathbb{P}(k_j|q_i) + \mathbb{P}(\Delta_j|q_i) \tag{6}$$

Where $\Delta_j = |x_i - x_j| + |y_i - y_j|$ is the spatial distance between event $i$ and $j$, $\mathbb{P}(k_j|q_i)$ is the posterior probability of getting key vector $k_j$ given the query $q_i$ and $\mathbb{P}(\Delta_j|q_i)$ is the posterior probability of event $j$ occuring at a spatial distance of $\Delta_j$ from the current event $i$, given the query $q_i$.

### 3.3.1 FIRST TERM OF EQ. 6

We start with Bayes' theorem to express $\mathbb{P}(k_j|q_i)$:

$$\mathbb{P}(k_j|q_i) = \frac{\mathbb{P}(q_i|k_j)\mathbb{P}(k_j)}{\mathbb{P}(q_i)} \tag{7}$$

Following distributions are assumed for $\mathbb{P}(q_i|k_j)$, $\mathbb{P}(k_j)$ and $\mathbb{P}(q_i)$,

* $\mathbb{P}(q_i|k_j) \sim \mathcal{N}(q_i|k_j, \sigma_j^2 \mathbf{I})$, where $\mathbf{I}$ is identity matrix
* $\mathbb{P}(k_j) = \pi_j$
* $\mathbb{P}(q_i) = \sum_{k=1}^{N} \mathbb{P}(q_i|k_k, \Delta_k)\mathbb{P}(k_k, \Delta_k)$, a Gaussian mixture model with $\mathbb{P}(k_k, \Delta_k) = \gamma_k$ and $\mathbb{P}(q_i|k_k, \Delta_k) \sim \mathcal{N}(q_i|k_k, \sigma_{q_k}^2 \mathbf{I})$

Substituting the above into Eq. 7,

$$\mathbb{P}(k_j|q_i) = \frac{\frac{\pi_j}{\sigma_j} \exp\left[-\frac{(q_i - k_j)^T(q_i - k_j)}{2\sigma_j^2}\right]}{\sum_{k=1}^{N} \frac{\gamma_k}{\sigma_{q_k}} \exp\left[-\frac{(q_i - k_k)^T(q_i - k_k)}{2\sigma_{q_k}^2}\right]} \tag{8}$$

Simplifying, we get,

$$\frac{\frac{\pi_j}{\sigma_j} \exp\left[-\frac{(\|q_i{}^2\| + \|k_j{}^2\|)}{2\sigma_j^2}\right] \exp\left(\frac{q_i^T k_j}{\sigma_j^2}\right)}{\sum_{k=1}^{N} \frac{\gamma_k}{\sigma_{q_k}} \exp\left[-\frac{(\|q_i{}^2\| + \|k_k{}^2\|)}{2\sigma_{q_k}^2}\right] \exp\left(\frac{q_i^T k_k}{\sigma_{q_k}^2}\right)} \tag{9}$$

Assuming $q_i$ and $k_j$ are normalized, $\mathbb{P}(k_j|q_i)$ turns out to be:

$$\frac{\frac{\pi_j}{\sigma_j}\exp\left[-\frac{1}{\sigma_j^2}\right]\exp\left(\frac{q_i^T k_j}{\sigma_j^2}\right)}{\sum_{k=1}^{N}\frac{\gamma_k}{\sigma_{q_k}}\exp\left[-\frac{1}{\sigma_{q_k}^2}\right]\exp\left(\frac{q_i^T k_k}{\sigma_{q_k}^2}\right)} \tag{10}$$

### 3.3.2 SECOND TERM OF EQ. 6

Using Bayes' theorem again:

$$\mathbb{P}(\Delta_j|q_i) = \frac{\mathbb{P}(q_i|\Delta_j)\mathbb{P}(\Delta_j)}{\mathbb{P}(q_i)} \tag{11}$$

We model $\mathbb{P}(q_i|\Delta_j)$ and $\mathbb{P}(\Delta_j)$ as follows,

* $\mathbb{P}(q_i|\Delta_j) \sim \mathcal{N}(q_i|\Delta_j q_i, \sigma_{\Delta_j}^2 \mathbf{I})$. This makes sure that it utilizes the information obtained from events that occur spatially far from the current event $e_i$.
* $\mathbb{P}(\Delta_j) = \beta_j$

Substituting the above into Eq. 11, we get,

$$\mathbb{P}(\Delta_j|q_i) = \frac{\frac{\beta_j}{\sigma_{\Delta_j}}\exp\left[-\frac{(q_i-\Delta_j q_i)^T(q_i-\Delta_j q_i)}{2\sigma_{\Delta_j}^2}\right]}{\sum_{k=1}^{N}\frac{\gamma_k}{\sigma_{q_k}}\exp\left[-\frac{(q_i-k_k)^T(q_i-k_k)}{2\sigma_{q_k}^2}\right]} \tag{12}$$

Reducing the terms $(q_i-\Delta_j q_i)^T(q_i-\Delta_j q_i)$ and $(q_i-k_k)^T(q_i-k_k)$ and incorporating the assumption that $q_i$ is normalized, $\mathbb{P}(\Delta_j|q_i)$ becomes,

$$\frac{\frac{\beta_j}{\sigma_{\Delta_j}}\exp\left[-\frac{\left(1+\Delta_j^2\right)}{2\sigma_{\Delta_j}^2}\right]\exp\left(\frac{\Delta_j}{\sigma_{\Delta_j}^2}\right)}{\sum_{k=1}^{N}\frac{\gamma_k}{\sigma_{q_k}}\exp\left[-\frac{1}{\sigma_{q_k}^2}\right]\exp\left(\frac{q_i^T k_k}{\sigma_{q_k}^2}\right)} \tag{13}$$

### 3.3.3 COMBINING FIRST AND SECOND TERM

Substituting Eq. 10 and 13 into Eq. 6, we get,

$$w_{ij} = \frac{\frac{\pi_j}{\sigma_j}\exp\left[-\frac{(1-q_i^T k_j)}{\sigma_j^2}\right] + \frac{\beta_j}{\sigma_{\Delta_j}}\exp\left[\frac{-\left(1-2\Delta_j+\Delta_j^2\right)}{2\sigma_{\Delta_j}^2}\right]}{\sum_{k=1}^{N}\frac{\gamma_k}{\sigma_{q_k}}\exp\left[-\frac{1}{\sigma_{q_k}^2}\right]\exp\left(\frac{q_i^T k_k}{\sigma_{q_k}^2}\right)} \tag{14}$$

## 4 EXPERIMENTS AND RESULTS

We evaluate whether polarity-driven self-supervised pretraining yields label-efficient representations for road segmentation, and whether the proposed probabilistic attention provides measurable gains over standard dot-product attention. Experiments address three core questions:

1. **Label efficiency:** Can PolFormer achieve competitive road segmentation with only a fraction of labeled data?

2. **Generalization:** Do event-only self-supervised representations transfer across datasets?

3. **Ablation:** How much does each component, self-supervision and probabilistic attention, contribute to performance?

| Method | Accuracy | mIOU | Method | Accuracy | mIOU |
|---|---|---|---|---|---|
| VID2E | 0.94 | 0.81 | DeepLabV3+ | 0.94 | 0.88 |
| EvSegNet | 0.95 | 0.79 | FCN | 0.89 | 0.77 |
| EvSegFormer | 0.95 | 0.85 | MobileNetV3 | 0.89 | 0.79 |
| LFD_Roadseg | 0.81 | 0.79 | PSPnet | 0.87 | 0.78 |
| SegFormer | 0.94 | 0.87 | DeepLabV3 | 0.87 | 0.77 |
| **Proposed**$_a$ | 0.90 | 0.73 | **Proposed**$_b$ | 0.93 | 0.81 |

Table 1: Comparison of PolFormer with state-of-the-art segmentation methods on DSEC-Semantic dataset. Other segmentation methods are fine-tuned for binary road segmentation tasks with all the labelled training samples (approximately million samples). PolFormer was initialized with self-supervised pre-trained weights and subsequently fine-tuned with 5.12k (**Proposed**$_a$) and 256k (**Proposed**$_b$) samples.

## 4.1 DATASETS

We benchmark on two standard driving datasets: DSEC-Semantic (24) and DDD17 (8). **DSEC-Semantic**: Proposed by (24). This is an extension of the DSEC semantic segmentation dataset made up of 53 driving sequences collected by a high standard frame-based camera and a high-resolution ($640 \times 480$) monochrome Prophesse Gen3.1 event camera. **DDD17**: The driving dataset DDD17 was recorded by the DAVIS346 event camera.

In this work, we have amalgamated non-road labels into a single class to transform the general semantic segmentation labels into road segmentation task. To simulate label scarcity, we vary labeled subsets from 5.12k to 256k samples, while leveraging large amounts of unlabeled events for pre-training. Based on empirical syudy, $N = 50$ was found to be optimum for the proposed PolFormer.

## 4.2 COMPARISON RESULTS ON DSEC-SEMANTIC

This section focuses on comparing the efficacy of self-supervised learning of PolFormer with against supervised baselines that aggregate events into frames for a window of every 50ms and apply conventional segmentation networks. These networks were trained on all labeled samples for each dataset on binary road segmentation task.

Table 1 summarizes the following results:

1. **Low-label regime**: With only 5.12k labeled samples, PolFormer achieves 0.90 accuracy and 0.73 mIoU, outperforming several fully supervised baselines trained on millions of labels.

2. **Moderate labels**: At 256k labels, PolFormer reaches 0.93 accuracy and 0.81 mIoU, matching or exceeding frame-based state-of-the-art.

These results highlight that polarity-driven pretraining allows PolFormer to scale down label requirements by orders of magnitude without sacrificing performance.

## 4.3 TRANSFER ACROSS DATASETS (DSEC TO DDD17)

In this section, we delve into evaluating transfer learning performance on the DDD17 dataset in fine-tuning mode. PolFormer base architecture was trained in self-supervised mode with the DSEC-Semantic dataset. Following this pre-training, a dense layer comprising 128 nodes and ReLU activation and a classification layer with 2 nodes are appended to the base architecture. During the fine-tuning stage, PolFormer was trained end-to-end using 256 labelled samples for 10 epochs. Optimization was carried out using AdamW optimizer with a learning rate set at 0.001.

The results (Table. 2) demonstrate that self-supervised PolFormer performs superior compared to existing supervised semantic segmentation architectures. Note that the proposed PolFormer was able to achieve state-of-the-art accuracy with limited labelled data of the DDD17 dataset. Competing baselines require full supervision to approach similar accuracy.

| Method | Accuracy | mIOU |
|---|---|---|
| VID2E (10) | 0.96 | 0.80 |
| EvSegNet (7) | 0.96 | 0.81 |
| EvSegFormer (25) | 0.97 | 0.82 |
| Mask2Former (29) | 0.92 | 0.87 |
| **Proposed** | 0.95 | 0.92 |

Table 2: Comparison of PolFormer with state-of-the-art segmentation methods on DDD17 dataset. PolFormer was initialized with self-supervised pre-trained weights (of the DVSEC dataset) and subsequently fine-tuned with 256k samples. Other segmentation methods were re-trained for the binary road segmentation task with all the labeled samples of the training data.

| # Labeled Samples (k) | Test-I | Test-II | Test-III |
|---|---|---|---|
| 5.12 | 0.76 | 0.78 | 0.72 |
| 128 | 0.89 | 0.93 | 0.78 |
| 5.12 | 0.91 | 0.93 | 0.84 |
| 128 | 0.9 | 0.93 | 0.83 |

Table 3: Analysis of PolFormer in terms of accuracy while initialized with pre-trained and random weights. Row 1 and 2: Random weights; Row 3 and 4: Pre-trained weights. There is notable difference between pre-trained and random weight initialisation, particularly when the labeled samples is scarce (5.12k)

This demonstrates that PolFormer learns generalizable event-native features rather than overfitting to dataset-specific distributions.

## 4.4 ABLATION STUDY

### 4.4.1 SELF-SUPERVISION VS. RANDOM INITIALIZATION

This section evaluates the performance of PolFormer by comparing the use of self-supervised pre-trained model initialization against random weight initialization. Pre-trained weights are generated by training the base network on the self-supervised task with randomly selected 102.4k unlabeled events. The base network is followed by a dense layer with ReLU activation and a classification layer. PolFormer is end-to-end fine-tuned and trained from scratch with 5.12k and 12.8k labeled samples. Comparison in terms of accuracy and mIOU is provided in tables 3 and 4. The results indicate a notable performance boost when employing pre-trained weights, particularly when the number of labeled samples is limited.

### 4.4.2 PROBABILISTIC VS. CONVENTIONAL ATTENTION

To assess the effectiveness of our proposed probabilistic attention, we compared it against the conventional scaled dot-product attention used in Transformer architectures. The probabilistic attention consistently outperformed the baseline in terms of accuracy (Table 5). This improvement can be attributed to its ability to integrate spatial distance, allowing the model to better exploit the inher-

| # Labeled Samples (k) | Test-I | Test-II | Test-III |
|---|---|---|---|
| 5.12 | 0.71 | 0.65 | 0.53 |
| 128 | 0.78 | 0.79 | 0.56 |
| 5.12 | 0.81 | 0.76 | 0.63 |
| 128 | 0.84 | 0.8 | 0.64 |

Table 4: Analysis of PolFormer in terms of mIOU while initialized with pre-trained and random weights. Row 1 and 2: Random weights; Row 3 and 4: Pre-trained weights. mIOU of PolFormer with pre-trained weights is greater than its random initialization version.

| Attention | Accuracy |
|---|---|
| Proposed | 0.93 |
| Dot Product | 0.92 |

Table 5: Comparison of probabilistic attention mechanism of PolFormer with conventional dot-product attention. Note the improved accuracy obtained with probabilistic attention mechanism

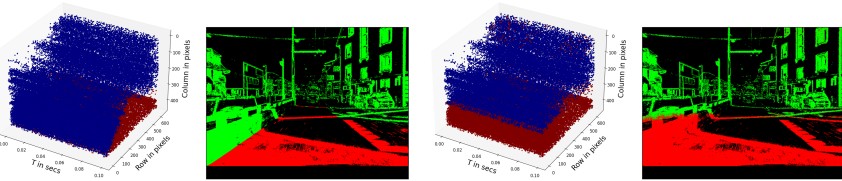

Figure 2: Visualization of input (col 1 and 2) and output (col 3 and 4) of PolFormer. Plots in col 1 and 3 show the 3D events, colored based on true and predicted label (red: road, blue: non-road). Plots in col 2 and 4 show the event frame created by populating each pixel with its recent event (Red: road, Green: non-road)

ent structure of event data. In contrast, standard dot-product attention treats all event interactions uniformly, which dilutes event-specific cues and leads to less robust representations. Notably, probabilistic attention demonstrates its suitability for large-scale event-based learning under label-limited conditions.

### 4.5 QUALITATIVE RESULTS

Figure. 2 shows the visual input and output of the propsoed method while fine tuned with a limited subset of 5.12k labeled samples. The visualization evidently illustrates the network's ability to predict road vs non-road segments even when trained on notably small set of labeled samples.

### 4.6 LIMITATIONS

The current study focuses on binary road vs. non-road segmentation without latency or throughput benchmarks, and relies on fixed event accumulation parameters ($N = 50$ based on empirical study) whose sensitivity is not extensively explored. Future work will extend to multi-class segmentation and dynamic accumulation strategies.

## 5 CONCLUSION

We introduced PolFormer, an event-native transformer that combines polarity-driven self-supervision with a probabilistic attention mechanism to tackle the label scarcity challenge in event-based road segmentation. PolFormer learns transferable representations directly from raw events, thereby preserving the sparsity and asynchrony that make event cameras unique. Experiments on DSEC-Semantic and DDD17 show that PolFormer achieves state-of-the-art segmentation with up to 200x fewer labels, while transferring effectively across datasets. Beyond road segmentation, this framework opens opportunities for applying event-native self-supervised learning to broader perception tasks such as multi-class segmentation, motion understanding, and 3D scene perception. Future work will extend PolFormer to multi-class and large-scale scenarios, explore adaptive event accumulation strategies. In summary, PolFormer demonstrates that high-level scene understanding is possible without sacrificing the unique efficiency of event sensing, paving the way for next-generation event-based learning systems.

## 6 ACKNOWLEDGEMENT

We acknowledge the use of Large Language Model (GPT) for help in grammar and phrasing of the paper.

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
