# OpenReview forum: "PolFormer: Event-Only Self-Supervision with Probabilistic Attention for Road Segmentation"
_ICLR.cc/2026/Conference — Submitted to ICLR 2026_

### Official Review · Reviewer_ZtoU · 2025-10-17

**Soundness:** 2
**Presentation:** 2
**Contribution:** 2
**Rating:** 2
**Confidence:** 5

**Summary:**

This paper proposed an event-native self-supervised transformer model that learns event-specific semantics directly from raw unlabeled event streams. The paper further proposed a probabilistic attention mechanism that outperforms standard dot-product attention on this modality. Experiments on DSEC-Semantic and DDD17 datasets demonstrate the effectiveness of the proposed method.

**Strengths:**

- This paper proposed an event-native self-supervised transformer model, which can learn rich event-specific semantics beneficial for road scene understanding.

- Experiments on DSEC-Semantic and DDD17 demonstrate the effectiveness of the proposed method.

**Weaknesses:**

- The proposed attention module should be compared against more event-relevant attention modules in the field to demonstrate the superiority of the proposed module. The differences and relations should also be discussed.
[] "Event-based semantic segmentation with posterior attention." IEEE Transactions on Image Processing 32 (2023): 1829-1842.
[] "EventASEG: An event-based asynchronous segmentation of road with likelihood attention." IEEE Robotics and Automation Letters 9.8 (2024): 6951-6958.

- More detailed ablations and parameter analyses should be conducted to help understand the design choices.

- The improvement is somewhat limited, as shown in the experiment tables, e.g., Table 1 and Table 5.

- Some event-based fusion methods could also be compared and discussed to help understand the performance gap between event-only methods and event-RGB fusion methods.

- The writing quality and presentation could be enhanced.

**Questions:**

- Would you consider making the source code publicly available to foster future research in this line?

- Would you consider reporting the per-class segmentation accuracy scores to help understand the benefits of the proposed method for different semantic classes?

---

### Official Review · Reviewer_qDkn · 2025-10-25

**Soundness:** 1
**Presentation:** 1
**Contribution:** 1
**Rating:** 0
**Confidence:** 4

**Summary:**

This paper proposes PolFormer, a Transformer-based framework for self-supervised learning on event-only data for road segmentation. The core idea is a pretext task based on event polarity to pre-train a model that learns "rich event-specific semantics". However, despite its initial idea, the paper suffers from serious issues, including a weak motivation, an unclear methodology, poor academic formatting, and insufficient experimental proof. These severe deficiencies undermine the paper's contributions and lead to a firm recommendation for rejection.

**Strengths:**

While the problem of unsupervised learning for event cameras is relevant, the paper's execution in terms of motivation, methodology, and experimental validation didn't present the strength of the paper.

**Weaknesses:**

1. Severe writing and formatting issues: The paper’s structure, formatting, and equation presentation deviate significantly from ICLR standards. The abstract and introduction contain multiple grammatical and logical errors (e.g., “exceptional dynamic range making them suitable for road segmentation”), duplicated numbering in contributions, and even raw markdown bullets in Section 3.3.1. The abstract and title capitalization also violate the ICLR style guide. This level of sloppiness makes the work nearly unreadable and suggests inadequate paper preparation.
2. Unclear and logically inconsistent methodology. The description of the proposed method is confusing and lacks internal logic.
- Line 102: It is unclear why “polarity entropy” should serve as an effective self-supervised signal for semantic segmentation rather than only motion or edge detection.
- The connection between polarity-driven pretraining and semantic feature learning is never established mathematically or conceptually.
- The probabilistic attention derivations are extremely verbose but unjustified; no intuition is given for why the additional Gaussian-based terms meaningfully model spatial locality.
3. Limited evaluation scope. All experiments are confined to binary road segmentation, which is not sufficient to demonstrate the richness or generality of the learned representations. For a paper claiming “rich event-specific semantics,” evaluations on more complex downstream tasks (multi-class segmentation, object detection, or motion estimation) are necessary.
- The probabilistic attention brings marginal gain (+1% accuracy), undermining its claimed contribution.
- The method is not fully self-supervised — it requires fine-tuning with labeled data, contradicting its “event-only self-supervision” claim.
- There is no experiment verifying robustness under different event densities or noise, nor any evidence of generalization to other event-based tasks.
4. Lack of motivation and insight. The authors reject the use of RGB supervision without justification, even though cross-modal supervision is well-established and effective. The motivation for an “event-only” approach seems ideological rather than technically grounded.
5. Overall impression. The manuscript reads as unfinished and incoherent, with questionable originality and very limited technical depth. It does not meet the basic clarity, rigor, or standards expected for ICLR.
6. Poor formatting and possible template violations. References do not follow ICLR citation style; the paper includes formatting artifacts (e.g., bullets, raw math, extra line breaks). The paper may not meet basic formatting standards and could be desk rejected.

**Questions:**

1. What is the theoretical motivation for using polarity entropy as a supervisory signal? How does this relate to semantic structure rather than just motion edges?
2. Why does the paper restrict evaluation to binary road segmentation? If the goal is semantic learning, why not evaluate on multi-class or object-level tasks?
3. The claimed “probabilistic attention” barely improves accuracy (0.93 vs 0.92). Can the authors justify its significance?
4. Why is fine-tuning with ground-truth labels necessary if the method is truly self-supervised?
5. Could the authors clarify missing definitions (e.g., how the probabilities in Eq. 6–14 are computed, what σ parameters represent, and how normalization is handled)?
6. How does this method compare to simpler SSL baselines or recent multimodal approaches using SAM-based representations?

---

### Official Review · Reviewer_uHnS · 2025-10-31

**Soundness:** 3
**Presentation:** 1
**Contribution:** 2
**Rating:** 2
**Confidence:** 4

**Summary:**

This paper proposes PolFormer, a self-supervised method designed for road segmentation training with only event-camera inputs. Instead of first transferring to RGB frames and then complete road segmentation, this paper proposes to train a transformer structure designed for event-camera data using a self-supervision scheme. The transformer is first self-supervised by polarity estimates derived from event data and then fine-tuned on a small fraction of labeled road segmentation data. A probabilistic attention mechanism is also proposed to make the attention modules attend more to regions with rich spatio-temporal information. Experiments show promising results of this method.

**Strengths:**

1. The motivation of the paper is clearly stated and easy to understand.
2. The experiment results are extensive, showing good real-life impacts.

**Weaknesses:**

1. Academic writing needs to be improved. There are some grammatical errors in the paper, and many sentences lack fluency or clarity. Examples include but are not limited to:
    - The references and citations do not appear to adhere to the ICLR format guidelines.
    - Line 018: "break" -> "breaks".
    - Line 054-064: Redundant numbering.
    - Table 1, Table 2: "Proposed" -> "PolFormer".

2. Many math symbols and expressions are not rigorous, which lead to confusions. Examples include but are not limited to:
    - Line 155 says $E$ is an "input tensor", but later Line 184 refers to $E$ as a "training dataset". Line 155 says $E$ has dimension $N\times 4$ (original data), but Line 212 says $E$ has dimension $N\times d_e$ (feature).
    - Eq (1) defines $p^+$ as the summation of all positive events, which I assume should be between 0 and $N$, but then Eq (2) uses $\log(1-p^+)$, which is not consistent with the previous definition.
    - Eq (5) needs to add $w_{ij} = $ on the left hand side.
    - Details about equation derivation (Sec 3.3.1-3.3.3) can be moved to appendix.

3. The introduction is very short and does not show many insights or understanding on this problem.

4. The experiment results on Line 362 states PolFormer is "matching or exceeding frame-based state-of-the-art", but Table 1 shows that PolFormer is worse than DeepLabV3+. It is uncertain whether DeepLabV3+ can also achieve comparable results with limited labels.

**Questions:**

N/A

---

### Official Review · Reviewer_ns52 · 2025-10-31

**Soundness:** 2
**Presentation:** 2
**Contribution:** 2
**Rating:** 4
**Confidence:** 2

**Summary:**

The paper tackles the problem of self-supervised representation learning from event-based camera, with downstream application of those representations to road segmentation. The authors introduce PolFormer - the model that learns from unlabeled event data using a self-supervised pretext task based on polarity entropy (i.e., how balanced positive and negative brightness changes are). It also introduces “probabilistic attention”, which adds spatial priors to the transformer’s attention mechanism. Overall, the authors show that the learned representatoins help with road segmentation with much fewer labels.

**Strengths:**

* The framing is novel. It is the first work to perform self-supervised representation learning on raw events. Also clever use of intrinsic signal (polarity) as self-supervision, which is elegant and data-efficient.

* Empirical results: solid generalization across datasets — rare in event-based vision. Also the label reduction factor (“200× fewer labels”) is impressive.

**Weaknesses:**

* Simplistic pretext task: The “entropy thresholding” binary task (Eq. 3) is very coarse. it is unlikely to encourage learning semantically aligned features beyond motion vs static regions. Thus, this representaiton is unlikely to help in more complex problems. There is also no analysis of whether learned representations transfer to non-motion-related tasks.

* Limited experiments: only binary road segmentation is tested, not general semantic segmentation.

* Another point is the absence of runtime or latency benchmarks, even though efficiency is a claimed motivation.

* The introduction of probabilistic attention with explicit spatial component seems like an overkill. The community has been using standard dot-product attention on images for a while now [a] and there is a way to implicitly model 2D proximity with standard sin/cos positional encosings.

[a] Dosovitsky et al., An image is worth 16x16 words: Transformers for image recognition at scale

**Questions:**

Could the authors demonstrate applicability of the learned features to more semantically challenging tasks, eg semantic segmentaiton?

---

### Meta-Review · Area_Chair_PHS4 · 2025-12-18

**Summary:**

This paper was reviewed by four experts in the field, and the reviews are mixed. The paper received scores of Marginal Reject (4), Reject (2), Strong Reject (0), and Reject (2).

This article proposes a new DNN PolFormer, a transformer-based DNN for self-supervised learning of representations from Event cameras. The paper focuses on road segmentation and autonomous driving. Rather than converting events data into RGB images, the proposed approach learns directly from  the raw data and uses a self-supervised task based on polarity entropy, which captures the balance between positive and negative brightness changes in the data. The model is first pre-trained in a self-supervised manner and then fine-tuned using a small amount of labeled road segmentation data.

Based on the reviews, I side with the reviewers recommending rejection. The authors are encouraged to carefully consider the reviewers’ comments, to improve the paper for submission elsewhere.

**Reviewer Concerns:**

The reviewers raised several major concerns that can be summarized as follows:
1. Clarity, Writing Quality, and Formatting:
Multiple reviewers criticize the form of the paper. There seem to be too many mistakes, and the mathematical equations are not always perfect; also the format of the paper is not conventional.
2. Methodological Motivation and Soundness:
The reviewers found that the relevance of the polarity entropy pretext task is not clear enough. Also, the probabilistic attention seems really hard to understand, maybe also due to the issues with the notations.
3. Limited Experimental Scope and Validation:
The experimental evaluation is restricted to binary road segmentation, which reviewers found insufficient to support claims of learning “rich event-specific semantics.” Several reviewers also noted missing or insufficient ablation studies.
4. Inconsistencies and Overstated Claims:
Some reviewers pointed out inconsistencies between claims in the text and reported results

**Reviewer Scores:**

The authors did not submit any response, so I don't think the reviewers would have increased their scores.

---

### Decision · Program_Chairs · 2026-01-26

Reject